# A Genome-Wide Alternative Splicing Analysis of *Gossypium arboreum* and *Gossypium raimondii* During Fiber Development

**DOI:** 10.3390/plants13192816

**Published:** 2024-10-08

**Authors:** Jianfeng Hao, Xingpeng Wen, Yuxian Zhu

**Affiliations:** 1College of Life Sciences, Wuhan University, Wuhan 430072, China; 18100333221@163.com (J.H.);; 2Institute for Advanced Studies, Wuhan University, Wuhan 430072, China; 3Hubei Hongshan Laboratory, Wuhan 430072, China; 4TaiKang Center for Life and Medical Sciences, Wuhan University, Wuhan 430072, China

**Keywords:** alternative splicing, *G. arboreum*, *G. raimondii*, transcriptome, fiber

## Abstract

Alternative splicing (AS) is a crucial post-transcriptional regulatory mechanism that contributes to proteome complexity and versatility in different plant species. However, detailed AS exploration in diploid cotton during fiber development has not been reported. In this study, we comparatively analyzed *G. arboreum* and *G. raimondii* AS events during fiber development using transcriptome data and identified 9690 and 7617 AS events that were distributed in 6483 and 4859 genes, respectively. *G. arboreum* had more AS genes and AS events than *G. raimondii*, and most AS genes were distributed at both ends of all 13 chromosomes in both diploid cotton species. Four major AS types, including IR, SE, A3SS, and A5SS, were all experimentally validated through RT-PCR assays. *G. arboreum* and *G. raimondii* had only 1888 AS genes in common, accounting for one-third and one-half of the total number of AS genes, respectively. Furthermore, we found a lysine-specific demethylase coding gene with a different AS mechanism in *G. arboreum* and *G. raimondii*, in which AS isoforms lacked part of a key conserved domain. Our findings may provide new directions for the discovery of functional genes involved in cotton species differentiation.

## 1. Introduction

Alternative splicing (AS) is a crucial post-transcriptional regulatory mechanism in eukaryotes that allows a single gene to produce multiple mRNA isoforms, which significantly contributes to transcriptome diversity and proteome complexity [1,2]. Since alternative splicing was first demonstrated in 1977 [3,4], along with technological advances and the popularity of high-throughput sequencing, AS has been comprehensively analyzed in many species on a genome-wide scale [5,6]. Over 90 and 61% of multi-exonic genes are alternatively spliced in humans [7] and *Arabidopsis thaliana* [8,9,10], and many isoforms produced by alternative splicing exhibit tissue or condition specificity [11], which has been linked to many human diseases [12,13]. However, not all AS events are functional; some isoforms produced by alternative splicing may contain a termination codon leading to degradation via the nonsense-mediated decay (NMD) pathway [14,15].

In plants, AS plays a crucial role in metabolism [16], temperature response, disease defense [17], immunity response [18], and osmotic stress response [19] and is regulated by histone modifications [20,21]. Most studies have identified and analyzed AS at the genomic level and found some functional AS genes, while others explained how alternative splicing contributes to plant development. For example, a wheat heat shock transcription factor, TaHSFA6e, produced a 14-amino acid peptide by AS on its C-terminal under heat stress, which enhanced the transcriptional activity of three downstream heat shock protein 70 (*TaHSP70*) genes [22]. In *Camellia sinensis*, when jasmonic acid (JA) was present, AS transcripts and *CsJAZ1* full-length transcripts interacted and formed heterodimers that stabilized the *CsJAZ1-CsMYC2* complexes, thereby repressing the transcription of four genes that act late in the flavan-3-ol biosynthetic pathway [23]. The AS transcript of *MaMYB16* was up-regulated during banana fruit ripening, competitively combined and formed non-functional heterodimers with full-length transcripts, decreased binding capacity with *MaDREB2*, and facilitated the activation of ripening-related genes, thereby promoting fruit ripening [24]. In *Oryza sativa*, *RLI1* alternative splicing produced a protein isoform without coiled-coil (CC) domains, enabling it to activate broader target genes that regulated brassinolide (BL) biosynthesis and signaling [25].

Cotton is an important economic crop worldwide and is used to produce both natural textile fiber and cottonseed oil [26,27]. Cotton is also an excellent model for studying genome polyploidization [28], cell elongation, and cell wall biosynthesis [29,30]. The *G. arboretum* (A2) and *G. raimondii* (D5) genomes, which are potential diploid ancestors of cultivated allotetraploid cotton species (*Gossypium hirsutum* and *Gossypium barbadense*), have been successfully sequenced and assembled [31,32]. *Gossypium arboreum* is an important cultivated diploid cotton species [33], whereas *G. raimondii* does not produce spinnable fiber. Although the *G. arboretum* genome is almost twice the size of *G. raimondii* [34], the number of annotated genes is similar, and >80% are orthologous [32]. Genome-wide association studies have identified many candidate genes associated with fiber traits [35,36,37,38,39]. Genome-wide analyses of the AS genes and AS events have been performed in *G. raimondii* [40], *G. arboretum* [30], *G. hirsutum* [41], and *G. barbadense* [42]. However, few systematic studies have revealed AS during cotton fiber development, and whether there are functional isoforms that regulate fiber development is unknown.

To visually compare fiber-associated AS, we collected transcriptomic data on the fiber developmental stages of cultivated diploid *G. arboreum* and *G. raimondii* (without spinnable fiber). Then, we conducted a comparative analysis of AS events and genes and chromosomal distribution and homologues in *G. arboreum* and *G. raimondii*. Histone modification-related AS genes were further analyzed, and a potential AS gene associated with fiber development was found. Our findings increase our knowledge of AS in different diploid cotton species and provide a new platform for studying the cotton fiber development mechanism.

## 2. Materials and Methods

### 2.1. Plant Materials

*G. arboretum* (Shixiya1) and *G. raimondii* (D5-3) were grown in a greenhouse at Wuhan University, Hubei, China. Then, 0, 5, 10, and 15 DPA (day post-anthesis) fibers and ovules were collected from *G. arboreum* and *G. raimondii*, immediately frozen in liquid nitrogen, and stored at −80 °C until RNA extraction.

### 2.2. RNA Extraction, cDNA Library Preparation, and RNA-seq

Fibers and ovules of the same weight from 0, 5, 10, and 15 DPA were mixed together and ground to powder with a grinder. Total RNA was extracted using a plant total RNA extraction kit (TIANGEN, DP441); after the removal of genomic DNA and ribosomal RNA, the mRNA was broken through the NEBNext^®^ RNA Fragmentation Buffer into 200–300 nt. The first strand of cDNA was then generated using random hexamer primers. Libraries were sequenced via Illumina platforms NovaSeq PE150. The low-quality reads and adapters were removed using Trimmomatic (version 0.36). The clean sequencing data of *G. arboreum* and *G. raimondii* have been uploaded to the NCBI Small Read Archive (SRA) and are available under the following accession numbers: SRR29754175 and SRR29754174.

### 2.3. Identification of AS Events

Genome sequences and annotations of *G. arboreum* (CRI) and *G. raimondii* (HAU) were downloaded from https://www.cottongen.org (accessed on 5 August 2023). Tophat (version 2.1.1) and Cufflinks (version 2.2.1) software [43] were used to align and assemble all of the paired-end clean reads. Transcripts with fragments per kilobase of exon per million fragments mapped (FPKM) <0.1 or ratios <10% of each gene were discarded [40,44]. Final transcript annotations were compared and merged with the reference genome using the cuffcompare tool (in Cufflink). We then used ASTALAVISTA v4.0 software [45] to identify the AS events. Four major AS types (IR, A3SS, A5SS and ES) were recognized, and the remainder were collectively grouped as Complex.

### 2.4. The Distribution of AS Genes and AS Events

To visualize the distribution of AS events and AS genes identified in *G. arboreum* and *G. raimondii* across their chromosomes, we calculated the density of genes and events based on their locations and visualized them using the Circos tool [46].

### 2.5. RT-PCR Verification of AS Events

Total RNA extracted from *G. arboreum* and *G. raimondii* were used to synthesize cDNA using a 1st Strand cDNA synthesis kit (Vazyme, R212-01) after the removal of genomic DNA. For each selected gene, primers were designed on the upstream and downstream of the splice site (Appendix A). RT-PCR reactions were performed under the following conditions: 95 °C for 5 min, followed by 30 cycles of 95 °C for 30 s, 57 °C for 30 s, and 72 °C for 60 s.

### 2.6. Homologous Gene Analysis and Conserved Domains Search

To investigate the homologous genes between *G. arboreum* and *G. raimondii,* we used a bidirectional blast based on protein sequences. To compare the different AS genes between *G. arboreum* and *G. raimondii*, we translated different transcripts into protein sequences for further analysis. We used the InterPro database (https://www.ebi.ac.uk/interpro/ (accessed on 5 May 2024)) to find conserved domains in protein sequences. The protein structure was predicted using alphafold (https://www.alphafold.ebi.ac.uk/ (accessed on 7 May 2024)).

## 3. Results

### 3.1. Statistical Analysis of AS Genes and AS Events

Cufflink and ASTALAVISTA were used to assemble the transcripts, estimate transcriptional expression, and identify AS events. In total, 27,598 and 28,123 annotated genes were expressed in *G. arboreum* and *G. raimondii*, which produced 44,711 and 41,563 gene isoforms (Table 1). Of all the genes expressed in *G. arboreum* and *G. raimondii*, 22,732 and 23,729 genes were multiexon. In our analysis, 9690 and 7617 AS events were identified in *G. arboreum* and *G. raimondii* and distributed in 6483 and 4859 genes (Appendix A).

Of the AS events identified, there were four major types, including retained introns (IRs), skipped exons (SEs), alternative 3′ splicing sites (A3SSs), and alternative 5′ splicing sites (A5SSs) accounting for 88.48 and 88.09% in *G. arboreum* and *G. raimondii*, respectively (Figure 1B). The remaining AS events were collectively grouped as “Complex”. In *G. raimondii*, 2453 IR events (32.2% of total events) from 1955 genes were identified, implying that most AS events were of this type (Figure 1A,C). In *G. arboreum*, 3039 A3SS events (31.36% of total events) from 2640 genes were the most abundant type. *G. arboreum* had more AS genes and AS events than *G. raimondii*.

All of the AS genes had at least one AS event. *G. arboreum* had more genes with one to five different AS events than *G. raimondii*, *G. raimondii* had more genes with over five AS events than *G. arboreum* (Figure 1D). An uncharacterized AS gene (Grai_02G007450) in *G. raimondii* had the most AS events, producing 14 from six different transcripts (Appendix A). *G. arboreum* and *G. raimondii* AS genes produced 1.49 and 1.57 splicing events on average.

### 3.2. Experimental Validation of Different AS Events Identified in G. arboreum

To validate AS events, four different AS event types in *G. arboreum* were randomly selected and verified by RT-PCR. The gene-structure models and amplified products represented different AS event types (IR, Figure 2A; SE, Figure 2B; A3SS, Figure 2C; A5SS, Figure 2D). As expected, two amplified products corresponding to the primary transcript and AS transcript were observed.

### 3.3. Distribution of AS Genes and AS Events on Chromosomes

To visualize the distribution of AS genes and AS events across *G. arboreum* and *G. raimondii* chromosomes, AS genes and AS event density were calculated. The *G. arboreum* A05 chromosome and *G. raimondii* D05 chromosome had the most AS genes and AS events, with 705 (1049 AS events) and 546 (852 AS events) AS genes, respectively (Figure 3 and Appendix A). Meanwhile, the *G. arboreum* A02 chromosome and *G. raimondii* D02 chromosome had the fewest AS genes, with 273 (395 AS events) and 262 (441 AS events) AS genes, respectively, and the *G. raimondii* D01 chromosome had the least AS events at 427. Generally, the distribution of AS genes was similar to expressed genes. Interestingly, almost all of the AS genes were mainly distributed at both ends of chromosomes, which was more obvious in *G. arboreum* (Figure 3).

### 3.4. G. arboreum and G. raimondii AS Gene Differences

*G. arboreum* and *G. raimondii* had almost the same number of genes, and most were homologous. To compare their genes, we used protein sequences to identify ortholog genes. We detected 27,598 and 28,123 expressed genes in *G. arboreum* and *G. raimondii*, respectively, from transcriptome data containing 24,983 pairs of *G. arboreum* and *G. raimondii* homologous genes (Figure 4A). Only 1888 AS genes were present in both *G. arboreum* and *G. raimondii,* 327 AS genes in *G. arboreum* and 397 AS genes in *G. raimondii* without expression in homologous genes. The remaining 4268 and 2574 AS genes were identified only in *G. arboreum* or *G. raimondii* (Figure 4B).

The AS gene exon numbers and lengths of the four majority AS events were statistically analyzed. IR genes had more exons than other expressed genes and were more pronounced in *G. arboretum* (Appendix A). The SE skipping lengths were similar in the two cotton species (Appendix A), while the retained intron lengths in *G. raimondii* were longer than those in *G. arboretum* (Appendix A). A3SS lengths were mainly below 40 bp, and A5SS lengths were more widely distributed (Appendix A).

### 3.5. A specific Transcription Factor Coding Gene JMJ25 Possesses a Distinct AS Mechanism in Both Gossypium Varieties

The gene’s protein sequence determines gene function, with those genes producing different transcripts through alternative splicing also creating different protein isoforms, which may have different functions. By comparing AS events between *G. arboreum* and *G. raimondii* homologous genes, we found a transcription factor jumonji (Jmjc) domain-containing protein coding gene named *JMJ25* (lysine-specific demethylase) with a different AS mechanism. In *G. arboreum* and *G. raimondii*, the gene produced three and two transcripts through alternative splicing (Figure 5A), which can translate into different proteins. All of the AS events were validated using RT-PCR, and different amplification product sizes were observed (Figure 5B). The results of the conserved domains showed that the *JMJ25* gene had three conserved domains, and all the AS sites in *G. arboreum* and *G. raimondii* were in the Jmjc-domain (Figure 5C), where Jmjc functions in a histone demethylation mechanism. The protein structure showed that the AS site was located in an alpha helix consisting of 10 amino acids at both ends (Figure 5D).

## 4. Discussion

High-throughput sequencing and high-quality genomes enable AS identification on a genome-wide scale. Although the total gene number in the genome and total number of genes expressed during fiber development were similar in the two diploid ancestors, *G. arboreum* and *G. raimondii*, AS genes and AS events identified in *G. arboreum* were greater than in *G. raimondii*. When comparing AS in homologous gene pairs between *G. arboreum* and *G. raimondii*, we found that over 88% of the expressed genes were the same, while > 61% of the AS genes were different. The same trends were observed in cultivated allotetraploid cotton (*Gossypium barbadense*) [42], so it appears that these differences were inherited from diploid ancestors and retained after polyploidization. *G. raimondii* and *G. arboreum* have similar genes but their fiber quality is very different. *G. arboreum* may produce more transcript isoforms by AS from a limited number of genes, increasing the diversity of gene transcription and the complexity of the proteome. In this study, the different AS genes between *G. arboreum* and *G. raimondii* may have played an important role in the development of fiber cells, especially in the stages of fiber initiation and elongation.

Among all of the AS events we identified in *G. arboreum* and *G. raimondii,* A3SS and IR were the most abundant types, accounting for 31.3 and 32.2%, respectively. The overall content of different AS types varied substantially between species; for example, IR is the most prevalent in plants and fungi, whereas ES is the most common in vertebrates [7] (38% in zebrafish, 42% in humans). These differences may be caused by transposable element (TE) insertions inducing changes in the branch point site distribution, which are important for IR [40]. During the development of cotton fiber cells, AS may produce alternative stop codons in mature mRNA sequences, which can be further processed or degraded via the NMD pathway [47,48] as required, ensuring efficient gene regulation in the process of fiber.

Gene expression involves transcription and translation, so the extent to which mRNA levels influence protein abundance and the effects in cases where this dependency breaks down remain topics of intense debate [49]. Normally, we determine whether a gene is up-regulated or down-regulated based on the level of transcription. There are so many dynamically changing genes during fiber development, especially during primary and secondary cell wall development [35,50]. However, few studies have reported that AS can change the encoded protein structure and function in cotton species. Comparing the differences in alternative splicing between two diploid cotton plants may provide new insights into the mechanisms of cotton fiber development, especially those upstream transcription factors and histone regulatory factors.

Histone modification is an important mechanism that mediates gene expression, growth, and development in plants and animals [8,9,10]. There are some modifications, such as Lys 4 trimethylation (H3K4me3) and histone H3 Lys K9 acetylation (H3K9ac), which are euchromatic marks that are often associated with active transcription, while other modifications, such as H3K9me2 and H3K27me3, are known as heterochromatic marks and are related to gene repression [49]. Moreover, histone modifications affect splicing outcomes by influencing the recruitment of splicing regulators via a chromatin-binding protein [21]. Additionally, H3K36me3 plays an important role in regulating AS and plant responses to high temperatures [20]. In allotetraploid cotton (*Gossypium hirsutum*), H3K4me3 levels are unequal in homologous gene pairs between A and D subgenomes [51]. We further compared the differences between *G. arboreum* and *G. raimondii* AS genes, especially transcription factors and including those related to histone modification. We found a *JMJ25* (lysine-specific demethylase) gene with a different AS mechanism in *G. arboreum* and *G. raimondii* whereby all transcripts can translate into proteins, and AS transcripts may have different protein activities that affect histone methylation levels. In Arabidopsis (*Arabidopsis thaliana*), the homologue *JMJ24* is a nuclear-localized Jmjc domain-containing protein that appears to regulate basal levels of transcription of silenced loci in part by controlling methylation in heterochromatic regions. Other homologues, *JMJ30* and *JMJ32*, mediate histone demethylation at the FLC site, constituting a balanced mechanism that controls flowering at elevated temperatures to prevent premature flowering [52]. In this study, we found that AS variants of *JMJ25* can change the activity of demethylases and may regulate the expression of downstream-associated genes, which suggested that *JMJ25* may be a potential gene associated with fiber development.

## 5. Conclusions

We used fiber development transcriptomic data to identify AS events in *G. arboreum* and *G. raimondii* at the genome level. The number of AS events in *G. arboreum* was significantly higher than *G. raimondii*. Furthermore, <39% of AS genes were identified in both *G. arboreum* and *G. raimondii*. There was a greater difference in the *G. arboreum* and *G. raimondii* transcriptome due to the presence of AS. Alternative splicing of lysine-specific demethylase *JMJ25* produced at least three- and two-protein isoforms with histone-modified demethylase activity that varied in *G. arboreum* and *G. raimondii* during fiber development.

*G. arboreum* and *G. raimondii* have similar genes but distinct differences in fiber traits. These ASs reflect an important transcriptional regulatory mechanism and potentially expand proteome diversity. The AS variants, especially those that differed between *G. arboreum* and *G. raimondii*, provide a new direction for studying the cotton fiber development mechanism.

## Figures and Tables

**Figure 1 plants-13-02816-f001:**
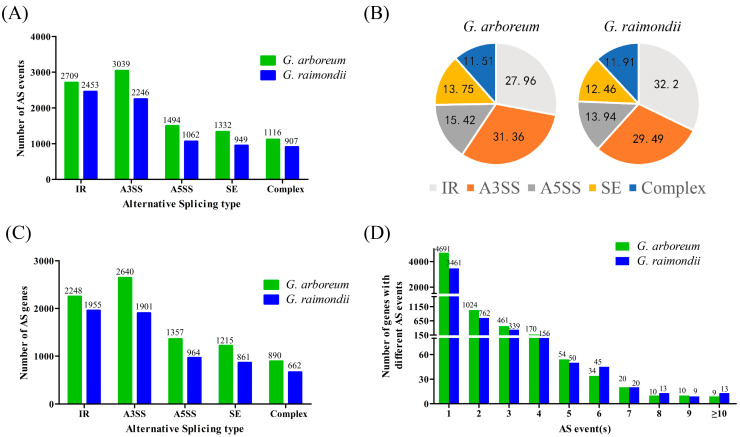
Statistics for different AS types and AS genes identified in *G. arboreum* and *G. raimondii*. Overview of different types of AS events: (**A**) number, (**B**) frequency, (**C**) gene number identified in *G. arboreum* and *G. raimondii*, (**D**) gene number distribution with different AS events.

**Figure 2 plants-13-02816-f002:**
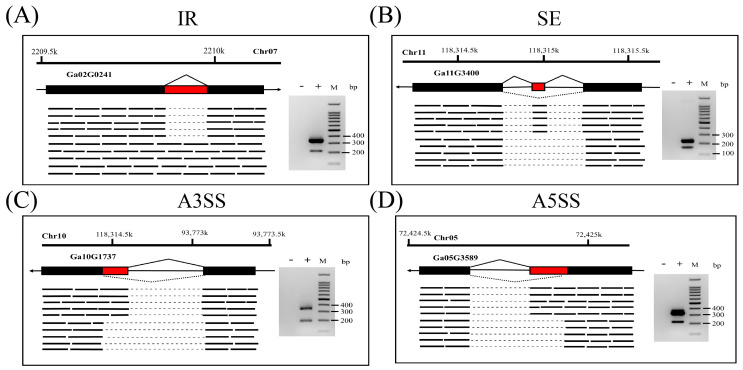
Experimental validation of the 4 main AS types in G. arboreum: (**A**) IR, (**B**) SE, (**C**) A3SS, and (**D**) A5SS. Black rectangles in the gene-structure models denote constitutive exons and red rectangles denote alternatively spliced exons; short lines under the gene-structure models identify the mapped reads, and dotted lines are genomic sequences that are not present in the RNA-seq data set. The gels show different-sized transcripts (PCR products) from the same primer pairs with cDNA templates; – represents the reverse transcriptase-negative PCR control.

**Figure 3 plants-13-02816-f003:**
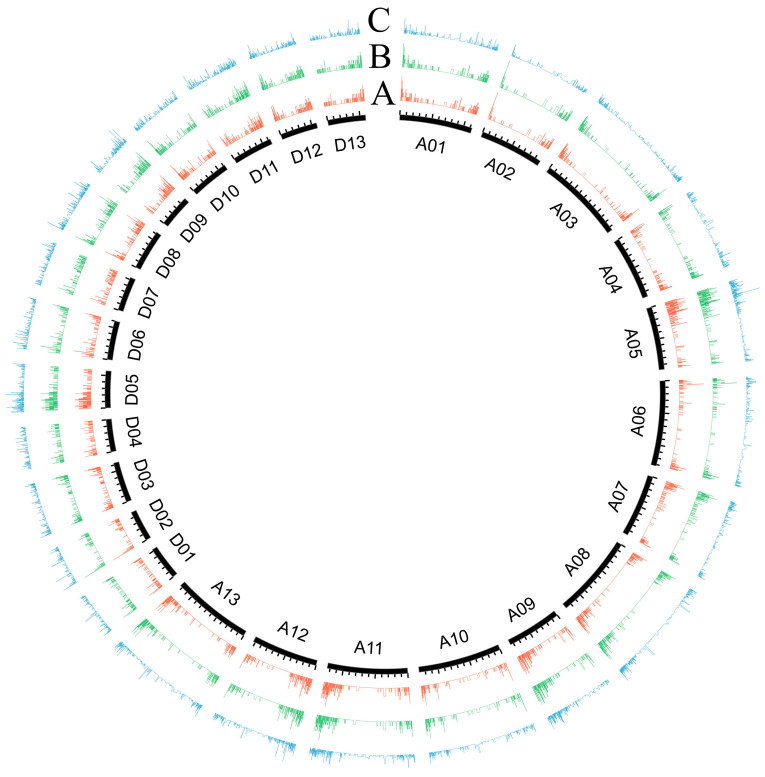
Distribution of AS genes and AS events across the *G. arboreum* (A genome) and *G. raimondii* (D genome) chromosomes. The densities of (**A**) all expressed genes, (**B**) AS genes, and (**C**) AS events.

**Figure 4 plants-13-02816-f004:**
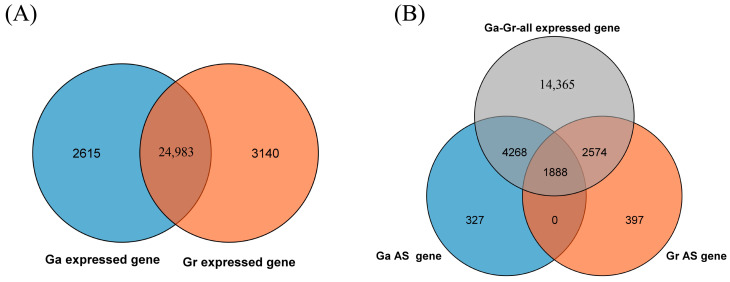
Venn diagram depicting the number of the homologous genes in *G. arboreum* and *G. raimondii*: (**A**) expressed genes in RNA-seq and (**B**) all expressed genes and AS genes from *G. arboreum* and *G. raimondii*.

**Figure 5 plants-13-02816-f005:**
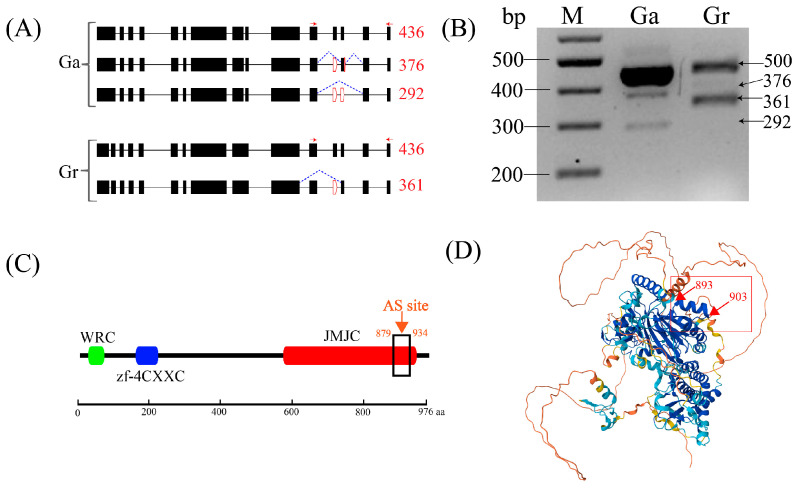
Different AS mechanisms of homologous genes between *G. arboreum* and *G. raimondii*. (**A**) represents the gene structure of the primary transcript and the AS transcript in *G. arboreum* (Ga13G1604) and *G. raimondii* (Grai_13G017400). The black boxes represent the exons; the black lines represent the introns; the dotted lines indicate the splicing results compared with the primary transcript; the red boxes represent the AS sites; the red arrows represent the forward primers and the reverse primers, and the red numbers represent the lengths of the forward primer to the reverse primer. (**B**) represents the RT-PCR validation of AS events; the gel bands show the DNA markers and the PCR results in *G. arboreum* and *G. raimondii*, which are amplified by the same primer, with its size (bp) indicated at the right. The conserved domains (**C**) and protein structure (**D**) of JMJ25 are also shown.

**Table 1 plants-13-02816-t001:** AS events and genes identified in *G. arboreum* and *G.raimondii*.

	Expressed Genes	Gene Isoforms	Expressed Multi-Exonic Genes	AS Genes	AS Events
*G. arboreum*	27,598	44,711	22,732	6483	9690
*G. raimondii*	28,123	41,563	23,729	4859	7617

## Data Availability

The transcriptome data of *G. arboreum* and *G. raimondii* has been uploaded to https://www.ncbi.nlm.nih.gov/sra/ (accessed on 4 July 2024) under the following accession numbers: SRR29754175 and SRR29754174.

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
