# Peer review of "A Genome-Wide Alternative Splicing Analysis of Gossypium arboreum and Gossypium raimondii During Fiber Development"

_plants, 2024, doi:10.3390/plants13192816_

Round 1

Reviewer 1 Report

Comments and Suggestions for Authors

The manuscript entitled “A genome-wide alternative splicing analysis in G. arboreum and G. raimondii during fiber development” by Hao et al. explore the mechanism of cotton fiber development by comparing the difference of alternative splicing (AS) between two diploid cotton ancestors for tetraploid cotton species. The differences of AS between G. arboreum and G. raimondii were analyzed in detail, and the AS genes that have functional differences were further analyzed. This is a novel topic and deserve to be published by the Plants journal. However, there are a few minor issues that need to be addressed before publication.

1.    In line 90, the software used in this study, they need to provide a detailed version number.

2.    “Transcripts with fragments per kilobase of exon per million fragments mapped (FPKM) <0.1 or ratios <10% of each gene were discarded.” Are there any references supporting this criterion?

3.    Figure 2 and Figure 3, the legends are reversed, check and revise it.

4.         Figure S1. Statistical tests should be performed to verify the significance.

Author Response

Response to Reviewer 1 Comments

Thank you very much for taking the time to review this manuscript. Please find the detailed responses below and the corresponding revisions in track changes in the re-submitted files.

Comments 1: In line 90, the software used in this study, they need to provide a detailed version number.

Response 1: Thank you for your valuable suggestion to improve this manuscript. We have added version number in the revised manuscript in lines 94-95.

Comments 2: “Transcripts with fragments per kilobase of exon per million fragments mapped (FPKM) <0.1 or ratios <10% of each gene were discarded.” Are there any references supporting this criterion?

Response 2: Thanks for your suggestion. Genes with FPKM less than 0.1 are often considered to have very low expression levels, possibly close to the detection limit of the assay, so we consider genes with FPKM less than 0.1 to be nearly non-expressed genes. ratios <10% of each gene were discarded to increase confidence in the identification of AS. We have added reference in the revised manuscript. (See lines 97)

References:

He, F., Wang, W., Rutter, W.B., Jordan, K.W., Ren, J., Taagen, E., DeWitt, N., Sehgal, D.,

Sukumaran, S., Dreisigacker, S., et al. (2022). Genomic variants affecting homoeologous gene

expression dosage contribute to agronomic trait variation in allopolyploid wheat. Nat Commun

13:826.

Grover, C.E., Gallagher, J.P., Szadkowski, E.P., Yoo, M.J., Flagel, L.E., and Wendel, J.F.

Homoeolog expression bias and expression level dominance in allopolypl oids. New Phytologist

196:966-971.

Comments 3: Figure 2 and Figure 3, the legends are reversed, check and revise it.

Response 3: Thanks for your suggestion. I apologize for any confusion caused. We have revised the

legends of Figure 2 and Figure 3. (lines 158-163 and lines 171-172)

Comments 4: Figure S1. Statistical tests should be performed to verify the significance.

Response 4: We appreciate your suggestion, we have performed statistical tests, See the revised version of Figure S1.

Reviewer 2 Report

Comments and Suggestions for Authors

The manuscript shows a significant study on the basic knowledge of transcriptome control in two species of the Gossypium genus. The information is relevant, but the major shortcoming of the document is the title promises research on fiber development, but there is nothing related to fiber development in the text. Without this evidence, the authors must rework the focus, and the manuscript will need extensive modification to deliver the promised information. I hope my comments will be helpful for the authors to fix the manuscript.

**Title:

-The title is about fiber development, but there is no evidence of fiber development. Please adjust the title after rewriting the document.

**Abstract: OK

**Keywords: OK

1. Introduction:

-The current flow is:

Alternative splicing as a significant post-transcriptional regulatory mechanism > Importance of alternative splicing on plants > Importance of diploid cotton to study potential alternative splicing on tetraploid cotton > Objective of this research

The flow is straightforward and clear. I see no need to change this section.

2. Materials and Methods

-The title should be numbered as “2”, not “2.2.”

2.1. Plant materials

-If possible, provide the accession code of the plants used in the study. Therefore, if somebody wants to know more information about the plants in the future, they can trace it back.

2.2. RNA extraction, cDNA library preparation, and RNA-seq:

-Please explain which part of the plant the RNA was extracted from. For example, did the authors extract from the fibers at 0, 4, 9, 16, 25, and 36 days post-anthesis? The metabolism of the cotton fibers is too fast, and a few days of difference may make a big difference in the extracted RNA.

2.3. Identification of AS events: OK

2.4. The distribution of AS genes and AS events: OK

2.5. RT-PCR verification of AS events: OK

2.6. Homologous gene analysis and conserved domains search: OK

3. Results

3.1. Statistical Analysis of AS genes and AS events

-The authors must explain from where the data was obtained. Please provide this information because the research loses much significance without it.

3.2. Distribution of AS genes and AS events on chromosomes:

-I suggest preparing graphs or tables per chromosome per species. For example, the authors could make a bar chart for G. arboreum, with 13 chromosomes. There would be a bar with the total genes and a smaller bar with the AS events, for each chromosome. If the authors agree with this suggestion but think it is too cumbersome for the main article, please do so in Supplementary Materials.

3.3. Experimental validation of different AS events identified in G. arboreum: OK

3.4. G. arboreum and G. raimondii AS gene differences: OK

3.5. A specific transcription factor coding gene JMJ25 possesses a distinct AS mechanism in both Gossypium varieties

-Please explain the significance of the JMJ25 for the fiber development. Is it a relevant gene for initiation, elongation, secondary cell wall thickening, or final maturation, for example?

4. Discussion

-The discussion is unrelated to the fiber development. The authors should discuss how the AS impacts the initiation, elongation, secondary cell wall thickening, or final maturation. This information is absent in the discussion, and the authors must rewrite the section.

5. Conclusions

-There is no evidence that the authors “used fiber development transcriptomic data to identify AS events in G. arboreum and G. raimondii at the genome level” (lines 266-267) because they did not explain from which development stage the data was collected. Even the example gente, JMJ25, is not directly related to the fiber development.

**Supplementary Materials: OK

**Author Contributions: OK

**Funding: OK

**Data Availability Statement: OK

**Conflicts of Interest: OK

**References

-Please capitalize all the nouns of the title or only the first noun. For example, the authors should write “Alternative splicing as a regulator of development and tissue identity” or “Alternative Splicing Plays a Critical Role in Maintaining Mineral Nutrient Homeostasis in Rice (Oryza sativa).” They cannot keep both systems at the same time in the References.

Author Response

Response to Reviewer 2 Comments

Thank you very much for taking the time to review this manuscript. Please find the detailed responses below and the corresponding revisions in track changes in the re-submitted files.

Comments 1: Materials and Methods,The title should be numbered as “2”, not “2.2.”

Response 1: Thank you so much for the advice. We have corrected the writing errors. (line 76)

Comments 2: 2.1. Plant materials, if possible, provide the accession code of the plants used in the study. Therefore, if somebody wants to know more information about the plants in the future, they can trace it back.

Response 2: Thanks for your suggestion. we have added the accession code of the plants used in this study in the revised manuscript. (line 78)

Comments 3: 2.2. RNA extraction, cDNA library preparation, and RNA-seq. Please explain which part of the plant the RNA was extracted from. For example, did the authors extract from the fibers at 0, 4, 9, 16, 25, and 36 days post-anthesis? The metabolism of the cotton fibers is too fast, and a few days of difference may make a big difference in the extracted RNA.

Response 3: I appreciate your advice. In the previous version of the manuscript, I did not describe the materials and methods clearly enough. I have rewritten the plant materials in this revision (lines 78-79 and lines 83-91).

Comments 4: 3.1. Statistical Analysis of AS genes and AS events. The authors must explain from where the data was obtained. Please provide this information because the research loses much significance without it.

Response 4. Thanks for your suggestion. I apologize for any confusion caused. In the revised manuscript, I rewrite and explained the data source in detail in “Materials and Methods” (lines 83-91). In this study, we used the transcriptome data of mixed fiber and ovule samples in G. arboreum and G. raimondii at the initiation and elongation stages of fiber development for subsequent analysis.

Comments 5: Distribution of AS genes and AS events on chromosomes. I suggest preparing graphs or tables per chromosome per species. For example, the authors could make a bar chart for G. arboreum, with 13 chromosomes. There would be a bar with the total genes and a smaller bar with the AS events, for each chromosome. If the authors agree with this suggestion but think it is too cumbersome for the main article, please do so in Supplementary Materials.

Response 5: Thank you for your valuable suggestion to improve this manuscript. This is a really good suggestion and we have added Figure S2 in the Supplementary Figures.

Comments 6: Please explain the significance of the JMJ25 for the fiber development. Is it a relevant gene for initiation, elongation, secondary cell wall thickening, or final maturation, for example?

Response 6: Thanks for your suggestion. The transcriptomic data is obtained from fibers and ovules initiation and elongation stages (0-15 DPA). JMJ25 is a lysine-specific demethylase and

the AS variants of JMJ25 are different between G. arboreum and G. raimondii. AS variants of JMJ25 can change the activity of demethylase and may regulate the expression of downstream genes. The specific function of JMJ25 for fiber development needs further study.

Comments 7: Discussion: the discussion is unrelated to the fiber development. The authors should discuss how the AS impacts the initiation, elongation, secondary cell wall thickening, or final maturation. This information is absent in the discussion, and the authors must rewrite the section.

Response 7: Thanks for your suggestion, I have rewritten the discussion and the role of AS related to the fiber development is added in the section (lines 227-231, lines 238-240, lines 244-250, 260-262, and lines 270-273).

Comments 8: Conclusions: There is no evidence that the authors “used fiber development transcriptomic data to identify AS events in G. arboreum and G. raimondii at the genome level” (lines 266-267) because they did not explain from which development stage the data was collected. Even the example gene, JMJ25, is not directly related to the fiber development.

Response 8: Thanks for your suggestion, I apologize for any confusion caused. I have rewritten the “Materials and Methods” and explained the data source in detail. G. arboreum and G. raimondii are diploid cotton ancestors of cultivated allotetraploid cotton species, as G. raimondii has no spinnable fiber, we collected mixed fiber and ovule samples from two diploid cotton at 0-15DPA (including fiber initiation and elongation period). In this study, we compare the difference of alternative splicing between two diploid cotton ancestors. The differences of AS between G. arboreum and G. raimondii were analyzed in detail, and the AS genes that have functional differences were further analyzed. RT-PCR validation and protein structure prediction revealed the AS variants of JMJ25 are different between G. arboreum and G. raimondii, we just infer JMJ25 might be associated with cotton fiber development in the discussion (lines 270-273)

Comments 9: References: Please capitalize all the nouns of the title or only the first noun. For example, the authors should write “Alternative splicing as a regulator of development and tissue identity” or “Alternative Splicing Plays a Critical Role in Maintaining Mineral Nutrient Homeostasis in Rice (Oryza sativa).” They cannot keep both systems at the same time in the References.

Response 9: I appreciate your advice. We have checked all the reference cited and corrected the errors in the revised manuscript.

Round 2

Reviewer 2 Report

Comments and Suggestions for Authors

The manuscript shows a significant study on the basic knowledge of transcriptome control in two species of the Gossypium genus. The information is relevant, and the authors clarified the text, showing evidence that alternative splicing has a role in fiber development. The research creates more questions than answers, and this document may be the seminal work for future research on alternative splicing for cotton fibers development. The authors must fix minor points, and I’m sure they can quickly fix them.

**Title:
-Please write “Gossypium arboreum” and “Gossypium raimondii.” The authors should not abbreviate the genus because it is the first time that the genus is written in the whole document. The species part must be in lowercase.

**Abstract: OK

**Keywords: OK

1. Introduction:
-The current flow is:
Alternative splicing as a significant post-transcriptional regulatory mechanism > Importance of alternative splicing on plants > Importance of cotton and the diploid cotton to study potential alternative splicing on tetraploid cotton > Objective of this research

The flow is straightforward and clear. I see no need to change this section.

2. Materials and Methods
2.1. Plant materials: OK
2.2. RNA extraction, cDNA library preparation, and RNA-seq: OK
2.3. Identification of AS events: OK
2.4. The distribution of AS genes and AS events: OK
2.5. RT-PCR verification of AS events: OK
2.6. Homologous gene analysis and conserved domains search: OK

3. Results
3.1. Statistical Analysis of AS genes and AS events: OK

3.2. Distribution of AS genes and AS events on chromosomes:
-Please reorganize the Figures 2 and 3. The citation of Figure 3 happens before Figure 2. Maybe just switching subsections 3.2. and 3.3. is enough to fix this.
3.3. Experimental validation of different AS events identified in G. arboreum:
-Please reorganize the Figures 2 and 3. The citation of Figure 3 happens before Figure 2. Maybe just switching subsections 3.2. and 3.3. is enough to fix this.
3.4. G. arboreum and G. raimondii AS gene differences: OK
3.5. A specific transcription factor coding gene JMJ25 possesses a distinct AS mechanism in both Gossypium varieties: OK

4. Discussion: OK

5. Conclusions: OK

**Supplementary Materials: OK
**Author Contributions: OK
**Funding: OK
**Data Availability Statement: OK
**Conflicts of Interest: OK
**References: OK

Author Response

    Thank you very much for your valuable comments on my manuscript. Please find the detailed responses below and the corresponding revisions in track changes in the re-submitted files.

Comments 1: Title: Please write “Gossypium arboreum” and “Gossypium raimondii.” The authors should not abbreviate the genus because it is the first time that the genus is written in the whole document. The species part must be in lowercase.
Response 1: Thank you so much for the advice. We have revised it according to  your suggestion. 

Comments 2: 3.2 and 3.3: Please reorganize the Figures 2 and 3. The citation of Figure 3 happens before Figure 2. Maybe just switching subsections 3.2. and 3.3. is enough to fix this.
Response 2:Thank you for your valuable suggestion to improve this manuscript. We have switched subsections 3.2. and 3.3 in the revised manuscript.